# Examining an Altruism-Eliciting Video Intervention to Increase COVID-19 Vaccine Intentions in Younger Adults: A Qualitative Assessment Using the Realistic Evaluation Framework

**DOI:** 10.3390/vaccines11030628

**Published:** 2023-03-11

**Authors:** Patricia Zhu, Ovidiu Tatar, Ben Haward, Veronica Steck, Gabrielle Griffin-Mathieu, Samara Perez, Ève Dubé, Gregory Zimet, Zeev Rosberger

**Affiliations:** 1Lady Davis Institute for Medical Research (LDI), Jewish General Hospital, Montreal, QC H3T 1E2, Canada; 2Research Center, Centre Hospitalier de l’Université de Montréal (CRCHUM), Montreal, QC H2X 0A9, Canada; 3Department of Psychology, McGill University, Montreal, QC H3A 1G1, Canada; 4Department of Oncology, McGill University, Montreal, QC H4A 3T2, Canada; 5Psychosocial Oncology Program, Cedars Cancer Center, McGill University Health Center, Montreal, QC H3A 3J1, Canada; 6Department of Anthropology, Laval University, Quebec, QC G1V 0A6, Canada; 7Department of Pediatrics, Indiana University School of Medicine, Indianapolis, IN 46202, USA; 8Department of Psychiatry, McGill University, Montreal, QC H3A 1A1, Canada

**Keywords:** COVID-19 vaccination, booster vaccination, altruism, qualitative, public health, health promotion, young adults

## Abstract

COVID-19 vaccine-induced immunity wanes over time, and with the emergence of new variants, additional “booster” doses have been recommended in Canada. However, booster vaccination uptake has remained low, particularly amongst younger adults aged 18–39. A previous study by our research team found that an altruism-eliciting video increased COVID-19 vaccination intentions. Using qualitative methods, the present study aims to: (1) identify the factors that influence vaccine decision-making in Canadian younger adults; (2) understand younger adults’ perceptions of an altruism-eliciting video designed to increase COVID-19 vaccine intentions; and (3) explore how the video can be improved and adapted to the current pandemic context. We conducted three focus groups online with participants who: (1) received at least one booster vaccine, (2) received the primary series without any boosters, or (3) were unvaccinated. We used deductive and inductive approaches to analyze data. Deductively, informed by the realist evaluation framework, we synthesized data around three main themes: context, mechanism, and intervention-specific suggestions. Within each main theme, we deductively created subthemes based on the health belief model (HBM). For quotes that could not be captured by these subthemes, additional themes were created inductively. We found multiple factors that could be important considerations in future messaging to increase vaccine acceptance, such as feeling empowered, fostering confidence in government and institutions, providing diverse (such as both altruism and individualism) messaging, and including concrete data (such as the prevalence of vulnerable individuals). These findings suggest targeted messaging tailored to these themes would be helpful to increase COVID-19 booster vaccination amongst younger adults.

## 1. Introduction

The ongoing COVID-19 pandemic has become the most serious pandemic in recent history, and variants of the SARS-CoV-2 virus continue to affect daily life. COVID-19 vaccines are estimated to have saved 19.8 million lives worldwide [1]. However, recent studies have shown that the initial COVID-19 vaccine-induced protection wanes over time [2], and is less effective against the new variants (e.g., Delta, Omicron) [3,4]. As a result, many countries, such as Canada, have recommended additional doses of COVID-19 vaccines (some of which have been updated to target new variants), often referred to generically as “booster shots” [5].

A major challenge faced in achieving optimal vaccination uptake is vaccine hesitancy, defined as a set of attitudes and beliefs that may lead to the delay or refusal of one or more vaccines despite their availability [6,7]. Vaccine fatigue, identified by Su, Cheshmehzangi [8] as “inertia or inaction towards vaccine information or instruction due to perceived burden and burnout”, can also contribute to inadequate vaccine uptake. Currently, in Canada, booster shot uptake for COVID-19 vaccines remains low, particularly amongst younger adults aged 18–39, at around 35–45% for the first booster and a mere 3–5% for the second booster [9]. Therefore, addressing booster shot vaccine hesitancy and fatigue amongst younger Canadians is crucial for achieving high vaccine uptake to contain the pandemic, limit its burden on the healthcare system, and manage endemic COVID-19 infections.

Several hypothetical, observational, and laboratory-game studies have suggested that prosocial behaviour and altruism were associated with vaccination intentions [10,11,12,13]. Prosocial behaviour can be defined as any behaviour that aims to generate welfare in others, while altruism is a subtype of prosocial behaviour that aims to increase others’ welfare without intentionally obtaining personal benefits [14]. In a randomized controlled trial (RCT), our research team previously evaluated the efficacy of an altruism-based video in increasing COVID-19 vaccine intentions amongst younger Canadians aged 20–39 [15]. We found a significant increase in binary (yes/no) vaccine intentions pre-to-post video intervention (4.8%), although there was no between-group difference in post-intervention when comparing to an active text control on non-vaccine-related preventive health measures. Importantly, we did find that participants in the video intervention arm who had not yet thought about or were undecided about the vaccine were more amenable to changing their intentions, as opposed to those who had already decided not to get vaccinated.

Using qualitative methods to better understand our previous study’s results and the association between prosocial motives (altruism) and COVID-19 vaccine intentions, the present study has three objectives: (1) to understand the factors that influence younger adults’ decisions to vaccinate; (2) to examine younger adults’ perceptions of an altruism-eliciting video designed to increase COVID-19 vaccine intentions; and (3) to explore how our video intervention could be adapted to meet the emerging challenges of booster doses.

## 2. Methods

### 2.1. Study Design and Ethics Statement

We used a qualitative study design and qualitative descriptive methodology to collect, analyze, synthesize, and interpret data [16,17]. Ethical approval was obtained from the Research Ethics Board of the CIUSSS West-Central Montreal (Project ID 2022-3133). Informed verbal consent was obtained through a Zoom video conferencing (Zoom) [18] meeting with a research assistant and recorded.

### 2.2. Participants and Setting

Participants who met the following eligibility criteria were included in the study: (1) Canadian resident; (2) between the ages of 18 and 39; and (3) had a working understanding of both English and French. Three focus groups were recruited, each including participants with varying COVID-19 vaccination statuses: (1) boosted, i.e., received the primary series of vaccinations against COVID-19 (two or more vaccine doses of Pfizer, Moderna, or AstraZeneca, or one dose of Johnson & Johnson) with at least one additional dose; (2) fully vaccinated, i.e., received the primary series without an additional dose; and (3) unvaccinated. These groups were chosen to reflect the major categories of COVID-19 vaccination reported by Health Canada [9]. Previous studies, including our RCT study that evaluated the efficacy of a video intervention, found that that individuals who had decided not to vaccinate were less likely to change their vaccine intentions than those who had not yet thought about or were undecided about vaccination [15,19].

### 2.3. Study Context

The focus groups took place between June and August 2022. At the time of the study, the National Advisory Committee on Immunization (NACI), the body responsible for national recommendations about vaccination in Canada, recommended an additional vaccine dose 6 months after the primary series for all adults 18 and over. Despite the primary series uptake being over 80% for those between 18 and 39 years old, only around 35–40% of younger adults had received at least one additional dose, and 16% remained unvaccinated against COVID-19 at the time of data collection [9]. Meanwhile, this age group continues to make up a large portion of infection cases (35%).

### 2.4. Recruitment

To recruit participants, we used purposeful sampling with an emphasis on heterogeneity between focus groups and homogeneity of vaccination statuses (i.e., boosted, fully vaccinated, and unvaccinated) within groups to more accurately obtain opinions amongst individuals with similar vaccine intentions [20]. Recruitment was carried out through social media platforms. Facebook study recruitment posts targeted university study recruitment groups (e.g., “McGill Studies for Cash”) and general study recruitment groups (e.g., “Survey and Focus Group for Canada and U.S.”). On Reddit, posts were made in Canadian groups (subreddits) related to COVID-19 (e.g., “r\CanadaCoronavirus”). Recruitment advertisements invited participants to participate in a focus group about “COVID-19 vaccination and public health messaging”.

### 2.5. Procedure

All focus groups were conducted using Zoom, with each focus group lasting between 75 and 90 min. Two moderators were present in all groups, the principal investigator (ZR), who had significant experience in qualitative research and discussion moderation, and a bilingual research assistant (GGM) who was responsible for taking field notes. Participants watched our previously developed altruism-based video in both English (https://tinyurl.com/2sbn4k9m) [21] and French (https://tinyurl.com/5xxctzb8) [22]. The video was developed around the idea that vaccination can provide a social benefit by protecting those who might not be able to vaccinate (e.g., at the time of video development, children), or might not develop an effective immune response (e.g., elderly, immunocompromised). A narrative structure was used given its recommendation for use in health behaviour interventions [23], and included three vignettes about vulnerable persons and the impact that others’ vaccination decisions had on their personal health. Further details about the content and the development of the video have been published elsewhere [15]. After this, the principal investigator conducted a semi-structured interview using the same interview guide for all three focus groups. Participants could provide responses in either English or French, and the research assistant provided clarification if needed in either language. Participants were compensated with a CAD 35 honorarium.

### 2.6. Interview Guide Development

The development of the interview guide and analysis were informed by a realist evaluation framework (REF) [24]. Understanding an intervention or program through a realistic evaluation lens requires the assessment of “how and why does this work and/or not work, for whom, to what extent, in what respects, in what circumstances and over what duration?” [25]. Pawson and Tilley [24] formalized this in context–mechanism–outcome configurations (CMO), which posit that understanding an intervention requires the identification of specific mechanisms triggered by it (e.g., altruistic motivations), the contextual factors that might modulate that mechanism (e.g., recommendations for booster vaccination), and how the interplay of these factors predicts varying outcomes such as vaccine intentions or uptake.

An interview guide was developed containing seven questions, two each related to ‘context’ and ‘outcome’, and three for ‘mechanism’. To identify contextual factors, questions sought views on the current context of the COVID-19 pandemic and vaccination, and how the situation has changed over time. Questions related to mechanism included how the presented altruism-based video might impact participants or others’ vaccination decision-making, both in the present context and retrospectively, how the message of the video could be improved, and what messages were perceived to be effective in the targeted age group. To understand the outcomes of the previous study [15], participants were informed that those who had “decided not” to receive the COVID-19 vaccine were unlikely to change their intentions after viewing the video, while those who were “unengaged” or “undecided” were more amenable to change. Participants were then asked to provide suggestions on what might have influenced participants to change or not change their vaccine decision-making stage after watching the video. The final interview guide is available in Appendix A: Interview Guide.

### 2.7. Analysis

An external transcription service (TranscribeMe) transcribed the focus group recordings, and accuracy in both English and French was confirmed by two members of the research team (BH, GGM). We imported the transcripts into NVivo v1.7.1 [26], which was used for data management.

To analyze transcripts, we used hybrid deductive–inductive thematic analysis. Deductively, we used the REF to create context, mechanism, and intervention-specific themes. Within the overarching themes, we created subthemes using the health belief model (HBM) [27], a widely used model of health behaviour, to frame our understanding of participants’ attitudes and beliefs about COVID-19 and the factors motivating vaccination. These subthemes included perceived susceptibility, perceived severity, perceived benefits (vaccine efficacy), and perceived barriers (harms of the vaccine). For quotes that did not reflect these subthemes, additional subthemes were developed inductively through an iterative process of reading the data multiple times and creating and refining codes. Starting with the boosted group, we used a sequential approach to data analysis in which four authors (VS, PZ, OT, BH) coded independently and then mediated coding collectively. Discrepancies were mediated by the senior researcher (ZR). The results of the first phase informed the analysis of the subsequent phases as we repeated the process using the thematic structure that resulted from the first transcript for those that followed. Using the thematic structure, four authors (VS, PZ, BH, OT) synthesized the information. The results were then discussed in meetings and validated by ZR.

## 3. Results

### 3.1. Focus Group Participants

In total, we recruited 18 participants. Seven participants were in the boosted group, seven in the fully vaccinated group, and four in the unvaccinated group. As per the inclusion criteria, participants were between the ages of 18 and 39, Canadian residents, and had a working understanding of English and French. To comply with the ethical requirements of our research ethics board (CIUSS West-Central Montreal), additional demographic information such as age, gender, and province of residence was not collected to protect the confidentiality of participants.

### 3.2. Thematic Structure

Contextual factors, mechanistic factors, and intervention-specific suggestions can all affect vaccine intentions. Context and mechanism factors can also influence each other, and they can both affect intervention-specific suggestions. See Figure 1 for the full thematic structure.

### 3.3. Context

#### 3.3.1. Attitudes and Beliefs towards COVID-19 Health Policies

Participants in the boosted group felt that there was less pressure to comply with health policies compared to earlier on in the pandemic, both from the government (less enforcement of health measures) and from the population (pandemic fatigue). Participants from both the boosted and unvaccinated groups believed that these health policies (for example, vaccine mandates, quarantine and isolation, random airport testing) were politically influenced, and some expressed doubt in their efficacy. In addition, participants from all three focus groups felt that the health policies were often inconsistent or were not well justified; they felt that the long-term strategy to overcome the pandemic was unclear, that the policies in place seemed disconnected from the direct methods that decrease viral transmission (e.g., getting fined for non-compliance toward health measures or taxes for the unvaccinated), and that these policies were an “overreaction” and extreme. Each quote below will refer to the group (i.e., “Boosted”, “Fully vaccinated”, “Unvaccinated” and participant (“P”) number.


*“The resolution seems to be, ‘Let’s get vaccinated every six months.’ But what about the next variant or the next pandemic?… I don’t think the solution is just vaccinate yourself out of fear every six months.” (Unvaccinated, P1)*


Participants from all focus groups felt that vaccination mandates were an infringement on personal rights and freedoms but ultimately increased vaccine uptake. Some expressed concerns that mandates could be extended to other domains.


*“So if we live in a society where I’m allowed to decide if I want to get the flu vaccine, I’m allowed to make a decision if I want to be on birth control. I’m allowed to make a decision to get an abortion. To me, this is no different. Where does it begin and where does it end?” (Unvaccinated, P3)*


#### 3.3.2. Mistrust in Government and Institutions

Unvaccinated participants shared the belief that the government was not being truthful about the side effects and efficacy of the COVID-19 vaccine, did not trust in pharmaceutical companies, and expressed general distrust in the government.


*“If you look at vaccine-induced myocarditis in younger people, I think the Canadian government says it’s—I think they were saying there’s Ontario did a study, I think it was one in 5000. But you look across the world, and it’s a lot more frequent in certain places. So I’d just like to see more transparency, and see these issues looked into more.” (Unvaccinated, P4)*



*“But the issue is six months from now, when that doesn’t happen, it just hurts trust. At this point, I’m not even sure who to trust. Right? Can you trust Trudeau [Canadian Prime Minister]? Can you trust Legault [Quebec Premier]? I can’t.” (Unvaccinated, P3)*


#### 3.3.3. Return to Normalcy

Boosted participants felt that restrictions were being lifted too fast, and life was returning to normal too quickly.


*“So they felt that it was good to lift the restrictions to live a more free life, but it was sort of neglecting that other people have different lives, different backgrounds, different family situations and whatnot, and so on. And yeah, I think people became a little self-centered in what they wanted and their desires of society, and they stopped realizing that everybody has a different situation.” (Boosted, P6)*


Participants from the boosted and fully vaccinated groups also wanted the pandemic to be over, and some believed that restrictions and mandates being lifted is a signal of the end of the pandemic.

#### 3.3.4. Perceptions about Media and Public Health Messaging

Regarding messaging content, participants from all focus groups felt that there was inadequate communication of scientific reasoning for policies, that messaging had not been transparent, and that messaging had been inconsistent.


*“What I would have liked to see during the pandemic that I wasn’t able to find myself, whether or not it was there, was a certain availability of… the science that was in between the direct academia and what I saw was potentially overly simplified messaging on the official government websites and maybe the newsclips in media that it wasn’t really getting… there was something kind of a middle ground, it was kind of missing, that I think I would have really enjoyed to get a better understanding of what actually was happening in the pandemic, and then I would be able to feel like I was making a more informed decision.” (Boosted, P4)*


Participants stated that the messaging had added to the panic, confusion, and fear of the pandemic, making it difficult for people to follow recommendations.


*“with the first messaging that you created, that was also at a time where we didn’t have as much information as we do now, and at that time as well there was—I don’t want to use the word ‘hysteria’, but there was a lot of panic and stress being felt along a lot of people who were, honestly, very scared of what was coming out.” (Fully vaccinated, P7)*


Participants in the boosted and fully vaccinated groups perceived that messaging intensity had decreased over time and felt that the messages had been delivered in outdated or inaccessible methods.


*“The general ads that roll on TV, I think that’s an antiquated way of relaying information or trying to convince people of doing things.” (Fully vaccinated, P3)*


### 3.4. Mechanism

#### 3.4.1. Perceived Susceptibility

Participants from all focus groups reported global (pandemic and preventive measures) determinants of susceptibility to COVID-19 infections. These included decreased perceived personal susceptibility because of adequate vaccine uptake among others in the population, which could then facilitate free riding (benefiting from protection against COVID-19 infection because others are vaccinated), and constant susceptibility to infection because of the emergence of new variants, the inability to eradicate the disease, and the sub-optimal efficacy of vaccines to limit transmission.


*“Given the fact that a significant amount of the population is already vaccinated, they don’t feel perhaps a need to get vaccinated now, because they feel as though they’re sort of immune to the challenges that are happening, yeah, I guess, as it relates to COVID-19.” (Boosted, P2)*


Participants also reported personal determinants of susceptibility, such as having higher susceptibility in closed and crowded spaces and for certain at-risk groups.


*“… but when I was forced to go back to school in September in person, that was when I was like “Okay. Now I am going into rooms with hundreds of people who I don’t know”, so that’s what pushed me more to get my vaccines.” (Fully vaccinated, P3)*


#### 3.4.2. Protecting Vulnerable Persons, Significant Others, the Community

Participants from the boosted and fully vaccinated focus groups indicated that protecting one’s family was one reason to get vaccinated, while participants from all groups acknowledged that protecting others was important, including those who are vulnerable (e.g., the elderly and immunocompromised), and the general community. 


*“and I’m actually an example, because my father was diagnosed with leukemia a week ago. So if I saw that video, it would be a push for me to go get the third and the fourth” (Fully vaccinated, P1)*


However, participants from all groups expressed that there has been a decrease in motivation to protect vulnerable persons, with some believing that this may be because the vaccine does not prevent transmission, and because the disease is impossible to eradicate.

#### 3.4.3. Perceived Vaccine Efficacy

Fully vaccinated and unvaccinated participants felt that vaccines were ineffective because of the emerging new variants, waning protection, not being as effective compared to other vaccines (e.g., chickenpox), and not having any impact on disease severity. As a result, unvaccinated participants believed that the inefficacy of vaccines could have implications in controlling the pandemic.


*“And also, if they were 95% effective, (the pandemic) would have been gone away. I’m determined that we’re all going to catch COVID, and we’re just giving each other false hope by vaccinating.” (Unvaccinated, P3)*


#### 3.4.4. Perceived Harms

Importantly, both fully vaccinated and unvaccinated participants expressed concerns and were uncertain about severe vaccine side effects, partly because of the novelty of the vaccine.


*“I think a lot of people are not okay with the not knowing… it can make one feel “Am I a lab rat?” (Fully vaccinated, P7)*



*“There’s so much learning that needs to be done around COVID, and long COVID, and the effects of COVID, and where the heck did COVID come from. Sorry, I don’t want to be your human guinea pig.” (Unvaccinated, P1)*


They felt that there was uncertainty about potential side effects and expressed that side effects will interfere with day-to-day responsibilities.


*“But in terms of the vaccine, how do I know what’s going to happen to me 10 years from now? If you guys don’t know what’s happening five days from now or a month from now, you can’t even tell me what’s going to happen 10 years from now.” (Unvaccinated, P2)*


Regardless of whether fully vaccinated or unvaccinated, participants also revealed that balancing the benefits of the vaccine to its side effects was important in deciding to get vaccinated or not.


*“I could see why people would not want to get the vaccine after having caught COVID, because why risk side effects for potentially no benefit and vaccine benefits post COVID” (Fully vaccinated, P3)*



*“…when I had gone to get a vaccine, and my decision not to do it was that I asked the person at the counter, I said, ‘If anything ends up happening and I get side effects, is there any recourse?’ And the answer was no. And that didn’t sit well” (Unvaccinated, P2)*


#### 3.4.5. Social Influence

Participants from all focus groups mentioned that vaccine decision-making can be influenced by family, friends, and the community around them. Specifically, one’s close circle can affect the perception of disease severity and vaccine benefits and harms.


*“Maybe personally, just around my circle, there’s people who have boosters, people who have double dose, some unvaccinated friends, and a lot of people around them got sick, and there didn’t really seemed to be a difference in terms of the severity of COVID when they caught it. And so, I guess, for me, personally, I didn’t think it was really necessary for me.” (Fully vaccinated, P4)*


Boosted and unvaccinated participants mentioned that social media represents an important information source, *“These are people, precisely, who will inform themselves using YouTube, and—‘I did my research’” (Boosted, P1),* and a way to exchange opinions with others who share similar political or health-related views.


*“I remember when people first started taking the shot, and there was TikTok videos of magnets. My cousin, it happened to her. She put a video on Facebook how after she took the shot, her arm was magnetized. So I believe my cousin, because it was working, there were magnets on her arm.” (Unvaccinated, P1)*


Those with anti-vaccine views could perceive being under pressure or marginalized by vaccine adopters, and embark on using social media that enables like-minded communications.


*“… when I did discuss with somebody in my entourage who was really against vaccinations, he didn’t believe in COVID, he—I think one of the things that he did mention is that every time he spoke to anybody, they really—antagonizing his views and made him even more, I think, want to go to an alternative media, where he felt there was a community with people like himself who had similar beliefs and where he felt more listened to.” (Boosted, P5)*


#### 3.4.6. Anti-vaccination Beliefs

Participants indicated that there is a distinct group of individuals who strongly adopt certain opinions and have low flexibility in changing their perspective. Amongst this group, some have always rejected vaccines (even before the COVID-19 pandemic) and changing their intentions would be difficult.

In addition, participants suggested that adopters of conspiracy theories are completely against vaccination.


*“I think people who are conspiracy theorists might feel like they’re really attacked all the time. They just don’t want to participate in that, and I think it makes them even more want to go against the grain and to not get the vaccinations, not wear masks” (Boosted, P5).*


Other beliefs included that the vaccination campaign has been pushed by the commercial interests of the pharmaceutical industry, and that the lack of trust in the government was fueled by the belief that it protects the commercial interests of pharmaceutical companies while neglecting the right to freedom of choice of the citizens. Another conspiracy included the belief that the forceful nature of COVID-19 health policies was a result of a predominance of men in positions of power who protected their own interests: *“Let’s say if COVID was a disease that had affected mostly women or mostly children, do you think the response would have been the same?” (Unvaccinated, P1).*

#### 3.4.7. Perceived Severity

Participants from all focus groups reported that perceiving COVID-19 symptoms being manageable (e.g., not serious, asymptomatic, having a full recovery) and having general good health status could reduce the urgency to receive COVID-19 vaccines, including booster shots:


*“Well, okay, a lot of my friends are getting it, and then after a couple days or maybe a week or two they’ve recovered.” (Boosted, P4).*


However, boosted and unvaccinated participants expressed that having high perceived severity for the COVID-19 disease, such as experiencing persistent unpleasant symptoms (e.g., loss of taste and smell), can elicit feelings of fear that could increase vaccine uptake in some individuals.


*“I haven’t seen anyone in a situation like this, this intense, but my sister, she caught it, and she still has effects, she hasn’t regained her sense of smell, taste. So it is quite intense. I would be very sad and depressed if I were in her situation.” (Boosted, P7)*


#### 3.4.8. Individualism

Participants from all focus groups expressed that being ego-centric could explain lower intentions to protect others, especially when vaccination is viewed as a personal rather than a collective responsibility. 


*“Well, I don’t have a relative with a major disease or I’m not having any particular health condition, so why would I get the vaccine?” (Boosted, P5).*


Fully vaccinated participants mentioned that having experienced mild COVID symptoms or perceiving the vaccine as less effective in preventing the transmission of the infection to others can dissuade individuals from adopting pro-social motives for vaccination.

In addition, individualistic persons could display lower flexibility in accepting vaccine recommendations.

“… *‘I’m the right one, I’m the smart one, you’ll see in the end, I was right’” (Boosted, P1),* or develop reactance to mandates, *“when you tell somebody you can’t do something, and they kind of dig their heels in, and kind of follow through with that.” (Unvaccinated, P4).*

To increase vaccine intentions, one participant suggested to think of one’s health as a prerequisite to preserve personal freedom.


*“…if you aren’t able to maintain your health and this does occur, there will be significant and long-lasting effects on freedom, if that’s your ultimate sort of value that you hold near and dear” (Boosted, P2)*


### 3.5. Intervention-Specific Suggestions

#### 3.5.1. Content

Fully vaccinated and boosted participants suggested that messages in the video should be diverse.


*“The messaging shouldn’t be like a one shoe fits all type of things, I feel like different people, different personalities, different levels of empathy, they have different reasons for why they get the vaccine, some people might be immunocompromised, some people want to protect their families, and so kind of a more personalized one-to-one messaging I think would be more effective in this kind of mass marketing type of messaging” (Fully vaccinated, P4)*


For example, messages should include the positive effects of vaccination on both healthy and vulnerable individuals, combine positive and negative stories related to health outcomes, and highlight that getting the vaccine is low-effort. Participants identified two main messages as useful: altruism and personal benefits. However, participants also noted that there is no universally effective message to promote vaccination, and while tailored messages may be effective, they would be difficult to implement.

Participants from all focus groups indicated the need to provide concrete information in the video. Many noted that viewing a video solely based on eliciting emotions can be *“emotional blackmail”, “manipulative”,* and *“like propaganda” (Boosted, P1; Fully vaccinated, P1; Fully vaccinated, P5; Unvaccinated, P1; Unvaccinated, P3; Unvaccinated, P4)*. Instead, participants requested that the video include more data related to the prevalence of vulnerable individuals, morbidity and mortality due to COVID-19 in vulnerable populations, side effects of the vaccine, and symptoms of COVID-19 that may be permanent or could take longer to recover, and be updated to the current pandemic context to avoid a repetition of old messages.


*“Just getting whole numbers. I’m not one for anecdotes, I don’t like these stories, I don’t like being lectured by an advertisement. I just want the numbers. I don’t know, it feels like there’s trickery going on when it’s these stories.” (Fully vaccinated, P3)*


Notably, participants also suggested that providing data related to vaccine efficacy could correct misconceptions about the vaccine (for example, that the disease offers better immunity than the vaccine).

Participants felt that messages should facilitate personal empowerment.


*“I think making it more personal and making people be able to understand the facts, but then also be able to voice their concerns in a way that they don’t feel, maybe, getting any pushback from society and to make it more of a private, personal choice” (Fully vaccinated, P7)*


Importantly, young adults would appreciate messages that facilitate collective empowerment and advocate for positive changes in society.


*“…or maybe figuring out a way to communicate how at an individual level and as a collective group people can create meaningful actions, so as to ensure that there is tangible change moving forward, I think would be a really powerful thing to do…” (Boosted, P2)*


#### 3.5.2. Design

Boosted and fully vaccinated participants suggested showing images of the damaging consequences of the COVID-19 disease.


*“… for example, is to show, perhaps, the significant and devastating effects of COVID, like to show graphic images of people lying in hospital beds, struggling to maintain their basic functions.” (Boosted, P2)*


Participants also proposed including images that could influence the vaccine decisional process, such as including healthcare professionals (e.g., doctors, nurses) telling of their experiences with COVID-19 or providing information related to the vaccine and for whom the vaccine is recommended. Some suggested that showing healthy young adults going about their daily activities and how that might impact the health of others may be more relatable than stories on protecting older, very young, or vulnerable individuals.

Participants from all focus groups expressed the need for visual representations to provide statistics and scientific information on vaccine efficacy, side effects of the vaccine, and how one’s everyday actions can impact others. 


*“‘Only 1% of people who have had the vaccine have suffered any serious side effects.’ Statistics. Numbers. Don’t come to me with feelings.” (Unvaccinated, P1)*


Related to perceptions about the video, several participants noted that the highly emotional video could convince some people, particularly those who are undecided about vaccination, to get vaccinated. Participants perceived the video as an advertisement (as it did not provide enough data that substantiate the efficacy and side effects), while others felt that the video was too childish (as it used animations) and felt lectured. However, the design of the video that showcased individuals of different ethnicities and occupations was appreciated.

Participants expressed that participating in an anonymous intervention could affect vaccine intentions because it creates a non-judgmental environment, which enables people to be more open-minded and accepting of new perspectives. In contrast, social desirability could also affect reported vaccine intentions.


*“It’s kind of hard to say to somebody’s face like, “I’m sorry, your ad didn’t actually help at all.” I’m not saying that that’s a fact. I’m just saying just the social element, I could see why somebody would say, ‘Yeah, you know, actually I do want to get vaccinated,’ but then, as soon as they leave the study, they’re like, ‘It didn’t change my mind at all.’” (Fully vaccinated, P3)*


Additional sample quotes for each theme and subtheme can be found in Appendix A: Themes and Quotes.

## 4. Discussion

Through conducting three focus groups with participants of diverse COVID-19 vaccination statuses, the present study aimed to explore factors that younger adults consider when deciding whether or not to receive COVID-19 vaccines, and to understand the influence of an altruism-eliciting video on their intentions to vaccinate.

Participants reported that they felt less pressure to comply with recommended preventive health measures, which might be attributable to pandemic fatigue, defined as a decrease in motivation to comply with health measures due to varying emotions, experiences, and perceptions over time [28]. Similarly, a study by Bodas, Kaim [29] suggested that booster shot vaccine hesitancy could be attributable to pandemic fatigue, while Cleofas and Oducado [30] found that lower pandemic fatigue was associated with higher likelihood of intentions to vaccinate. Intolerance of uncertainty has been found to be a predictor of pandemic fatigue [31]; this has also been highlighted by our focus group participants who expressed their frustrations toward the unclear rationale for health policies and uncertain long-term pandemic mitigation strategies, as well as the inconsistency in public health messaging. Similarly, a focus group study with participants from Alberta, Canada [32] found that public health messaging was perceived as conflicting and inconsistent. To minimize pandemic fatigue, preventive health behaviours, including vaccination, should be well justified and clearly communicated.

Our synthesis aligns with previous research showing that gain- and loss-framed messages could be effective in promoting COVID-19 vaccination [33,34,35,36]. For example, loss-framed messages could emphasize the severe consequences of COVID-19 that could be a result of non-vaccination, such as the persistence of post-viral infection symptoms (long COVID-19) [37], which include over 50 long-term effects [38]. A message related to the long-term and potentially serious symptoms of COVID-19 (as an example of a loss-framed message) could highlight the importance of loss of personal freedom, as having serious symptoms would hinder one’s ability to act freely. Meanwhile, gain-framed messages that emphasize the benefits of vaccination, including the vaccine’s ability to minimize risks of hospitalization and mortality, could also encourage vaccine acceptance.

Participants in our study expressed concerns toward the severity of potential side effects, the uncertainty of side effects, and the novelty of the vaccine. This reflects the current literature, which has found positive associations between perceived vaccine harms (safety, side effects, novelty of the vaccine) and vaccine hesitancy [39,40,41,42,43]. Many of the perceived harms of the vaccine, especially regarding its safety and efficacy, stem from anecdotal stories from people who experienced these effects. Due to availability heuristics, the widespread narratives of alleged side effects will often result in generalization of these side effects, despite their actual incidence rate [44]. In addition, “individuals prefer to know *how* consequences might be if they do occur, rather than *how likely* a consequence is to occur” [44]. As a result, emotional stories are often difficult to counteract by only providing scientific evidence, as social media activity of individuals is higher than of health institutions. Instead, efforts should be made to encourage individuals who have had positive experiences with the vaccine or who did not vaccinate and experienced serious COVID symptoms to share their personal stories.

Participants acknowledged social media to be both an information source and a place to exchange opinions with those who have similar health or political views. Social media has previously been identified as a method for rapid information spread, including content that promotes anti-vaccination [45]. A qualitative study by Lockyer, Islam [46] found that negative perceptions of COVID-19 vaccines on social media can create confusion and affect vaccine decision-making. As individuals who are mis- or uninformed are most likely to spread COVID-19 misinformation online [47], nudging them to consider the accuracy of the information they read online could be an effective method to limit the spread of misinformation [48,49].

Altruism was an important recurrent theme identified by our study participants that could positively influence vaccine intentions, particularly for individuals who have not yet thought about or are undecided about vaccination. Many studies found that increased altruism was associated with increased COVID-19 vaccine intentions both directly [50,51,52] and indirectly [53]. In addition to our RCT study, an experimental study by Rieger [54] concluded that altruistic messaging was most effective in increasing COVID-19 vaccine intentions. Specifically regarding younger adults, our research team had previously found that a higher preference for altruistic motives predicted vaccine acceptance [55], while another Japan-based study identified prosocial traits as a major influence on vaccine acceptance [56]. Contrasting altruism, participants also cited individualistic reasons to get vaccinated. A study by Gong, Tang [33] found that when comparing gain-framed, loss-framed, and altruistic messaging, loss-framed messaging was more effective in increasing COVID-19 intentions. Future behavioural interventions aimed to change vaccine intentions should incorporate one or both messages, as their relevance was identified by our study participants and by the existing literature.

As seen from our focus group discussions, both individualistic and altruistic messages could influence vaccine intentions; thus, messaging to promote vaccination should be diverse, and tailored messages should be created for people with different needs. One way to reach people with targeted messages could be through using chatbots, which are computer programs that use natural language to communicate and interact with their users [57]. Several studies have found that chatbots significantly increased people’s intentions to vaccinate [58,59,60,61]. Providing access to different tailored messages would allow individuals to be more informed without feeling pressured, which could in turn facilitate the empowerment of personal choice amongst younger adults. A study by John, McAndrews [62] found that nudges that encouraged people to reflect on their personal health decisions increased vaccine intentions. Behavioural nudging, or nudging, suggests that promoting positive impacts of a behaviour without changing incentives or mandating actions can significantly affect the behaviour [63]. Appealing to emotional affect could be another method of nudging [64]. This reflects the idea that our focus group participants suggested, for which our original, highly emotional video may persuade participants who had not yet thought about or were undecided about receiving the vaccine. Interestingly, this also paralleled our RCT study findings in which participants in the video intervention arm who were unengaged (had not yet thought about) or undecided about taking the COVID-19 vaccine were more amenable to change after viewing the video. Thus, creating emotion-invoking content could help to promote vaccination amongst those in earlier stages of vaccine decision-making. However, some participants also felt that an emotional message, or even any message, would not work for those who are strongly against vaccination. This is in line with studies that found that “rigid” and “strongly” hesitant individuals are less likely to change their decision [19,65], and more research on creating interventions for this group is needed.

Participants from all focus groups felt that COVID-19 health policies were an infringement on personal freedom and human rights, with some even expressing concern that mandates could extend to other domains. While mandatory vaccination successfully increased vaccine uptake, it could potentially exert significant negative effects on personal psychology (e.g., developing reactance), politics [e.g., diminishing civil liberties), socioeconomics (e.g., increasing health disparities and inequalities), and trust in science and public health [66]. Instead, as voiced by our study participants, people want to make decisions for themselves, to feel empowered through the freedom of choice. One way to empower people to make decisions themselves could be through the use of decision aids [67]. These are tools aiming to guide individuals through the vaccination decision-making process by providing the probabilities, risks, and benefits of different options. Ultimately, offering the necessary resources to allow individuals to make their own decisions could preserve public trust in not only government and health institutions, but also science.

Notably, all participants in the unvaccinated focus group expressed mistrust in the government or institutions. Previous studies have shown that higher trust in the government and information from the government were associated with increased vaccine acceptance [68,69,70,71], while lack of confidence in state institutions was associated with COVID-19 mortality [72]. A global study by Lazarus, Wyka [73] from 23 countries worldwide identified decreased trust in science as a consistent correlate of vaccine hesitancy. Thus, fostering trust in the government, institutions, and science is critical for engaging the public in complying by preventive health recommendations, including vaccination. Hyland-Wood, Gardner [74] highlighted ten ways to promote trust amidst the COVID-19 pandemic, which included providing clear messaging, involving healthcare professionals and medical experts, remaining honest and transparent, empowering people to act, and countering mis- and disinformation; many of these sentiments were echoed by our focus group participants. Furthermore, listening to public opinions and creating communication campaigns accordingly could also build confidence [75].

With respect to video-related comments and suggestions, participants acknowledged that the inclusivity of the video made the video more relatable, and could persuade some people to receive a vaccine. Participants also appreciated the positive and negative stories; however, more data and concrete information, for example the prevalence of Canadians who are vulnerable or at-risk, should also be provided. Additionally, participants mentioned that the anonymity aspect of the online RCT study enabled individuals to answer questions in a non-judgmental environment, which could facilitate people to be more open-minded to new perspectives. Nevertheless, participants recognized that social desirability could affect reported vaccine intentions.

### Study Strengths and Limitations

We used theoretical frameworks [especially the CMO framework] to inform our synthesis, which is useful in studies using a sequential design to adapt and refine interventions. The use of the HBM facilitated the deductive analysis and resonated with factors perceived as important by young adults in the vaccine decisional process. Including three distinct groups of individuals based on their vaccination status allowed us to capture diverse opinions and provide a comprehensive synthesis of factors that influence vaccine intentions in the young adult population [76].

As we collected data in the summer of 2022, the pandemic context and health policies may have changed over time and could influence younger adults’ intentions to vaccinate. Because we only conducted three focus groups, we do not claim to have reached data saturation, especially for unvaccinated individuals who were reluctant to participate. Recruiting more unvaccinated participants could have allowed us to collect a broader range of opinions. Future studies should consider the discriminatory attitudes toward unvaccinated individuals as they can be as high or even higher than discrimination towards immigrants, ex-convicts, and drug addicts [77]. As a result, this can pose as a serious impediment to participant recruitment. More research is needed for an in-depth understanding of factors that influence vaccine intention, especially for “rigid” vaccine-hesitant individuals whose attitudes and beliefs are difficult to change [19]. Self-selection bias [78] may limit the generalizability of opinions expressed in this study as participants were informed they would be participating in a study about COVID-19 vaccination. Bias may also have been introduced by the fact that the same researcher (ZR) moderated all focus groups [76].

## 5. Conclusions

Building on a growing body of literature examining the impact of message framing on COVID-19 vaccine intentions and uptake, this study provides a unique and important qualitative insight into the factors that younger Canadian adults consider when making a decision regarding COVID-19 vaccination, and provides suggestions for future interventions. Notably, we found that increasing “pandemic fatigue”, perceived severity of vaccine side effects, and rigid, hesitant attitudes might underly vaccine hesitancy in this group, and suggest that the empowerment of decision-making, promotion of public confidence in government and institutions, and the integration of concrete data in messaging could be favorable strategies to increase vaccine acceptance. While participants acknowledged the importance of altruism in COVID-19 vaccine decision-making, they also stressed the need for diverse and targeted messaging. Our findings can inform public health authorities in creating targeted messaging in the context of additional and possibly regular booster vaccination, and in preparation for future outbreaks or pandemics in which young adults might be central to disease transmission, burden, and mortality.

## Figures and Tables

**Figure 1 vaccines-11-00628-f001:**
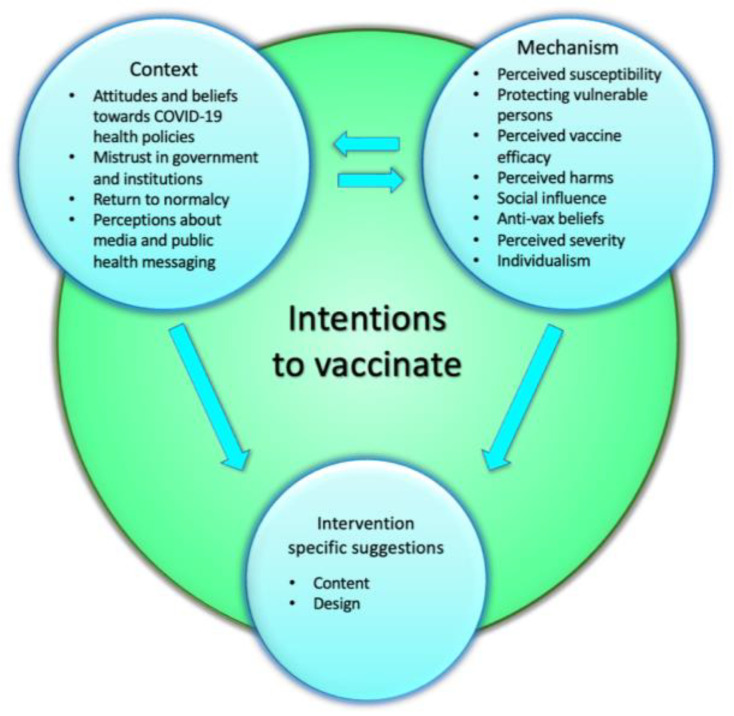
CMO Thematic Structure.

## Data Availability

The data used in this study will not be published in a publicly available repository, in accordance with ethical requirements. The data will be available from the senior author (ZR) upon reasonable request, and upon agreement of confidentiality and data use policies provisioned by the primary institution.

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
