# Peer review of "Examining an Altruism-Eliciting Video Intervention to Increase COVID-19 Vaccine Intentions in Younger Adults: A Qualitative Assessment Using the Realistic Evaluation Framework"

_vaccines, 2023, doi:10.3390/vaccines11030628_

Round 1

Reviewer 1 Report

This paper reports on a qualitative focus-group study of the impacts of an altruism-eliciting video on COVID-19 vaccination intentions. The paper is well-written, the study design and analysis follow state-of-the-art research protocols, and the findings are interesting and useful.

The one suggestion I have for revision of the manuscript is that the text of the paper in the Study strengths and limitations should include mention of the relatively limited sizes of the focus groups and the use of the remote access Zoom tool for the group interactions. It is not anticipated that larger, in-person focus groups would lead to greatly different views, but it would be good to study this in future sessions. 

Reviewer 2 Report

This paper is an intresting piece of work , using qualitative research methodology  on the field of Covid-19 vaccine.

There are a few minors questions

Please define or explain what "prosocial behavior" is so that unfamiliar readers can understand it.

Line 124 change  omit $ , because is redundant (CAD means Canadian Dollars) see Canada Government web pages.

https://www.btb.termiumplus.gc.ca/tcdnstyl-chap?lang=eng&lettr=indx_catlog&info0=5.11&info1=5.26

Please provide a linkt to the video that was used. (or include it as supplementary material) that will allow a better understanding of the context of the study.

Reviewer 3 Report

I have reviewed  very interesting manuscript with important findings.  Overall, the manuscript is very well-written; The rich value of qualitative study is the deep discussion from comparing result obtained from several data collection methods as various sources. Just simple question, do you use the triangulation process in validating process? I think you can explain that in the paper. 
